# Sensory Analysis of a Processed Food Intended for Vitamin A Supplementation

**DOI:** 10.3390/foods9020232

**Published:** 2020-02-21

**Authors:** Xiaoyu Zhang, George A. Cavender, Kristina R. Lewandowski, Ginnefer O. Cox, Chad M. Paton

**Affiliations:** 1Department of Food Science and Technology, University of Georgia, Athens, GA 30602, USA; Xiaoyu.Zhang1@uga.edu (X.Z.); cavender@uga.edu (G.A.C.); Kristina.Lewandowski@uga.edu (K.R.L.); 2Department of Foods and Nutrition, University of Georgia, Athens, GA 30602, USA; gocox@uga.edu

**Keywords:** β-carotene, vitamin A deficiency, 9-point hedonic test, product acceptability

## Abstract

Provitamin A and pre-formed vitamin A compounds are essential micronutrients for humans. However, vitamin A deficiency (VAD) affects the health status of nearly 50% of populations in Southeast Asia and sub-Saharan Africa and is especially pronounced in preschool children and pregnant women. The objective of this research was to determine an acceptable flavor/ingredient combination to produce a palatable food product that incorporates sweet potatoes, peanut paste, and chickpeas. We sought to determine the acceptability of the three product formulations and to determine the influence of demographic data on ratings for the sensory attributes of each sample. To address VAD issues, three formulations of a product incorporating sweet potato puree (to increase β-carotene content), pure peanut butter (to provide fat for β-carotene absorption), and chickpeas (to provide a complete protein source), were developed: (1) an unflavored control, and two formulations with added natural seasonings: (2) curry-flavored, and (3) pumpkin spice-flavored. Sensory analysis of the three products showed that the curry-flavored product received the highest acceptability in terms of overall liking, flavor, texture, and appearance (*p* < 0.001). Since the demographic effect was not statistically significant (*p* > 0.05), it is highly likely that the curry-flavored product can be implemented in other countries or areas with high acceptability.

## 1. Introduction

Due to insufficient intake of either pre-formed vitamin A or provitamin A carotenoids, more than 2 billion people are at risk of vitamin A deficiency (VAD) worldwide. In particular, children between the ages of 6 months–5 years and pregnant and/or lactating women are at the highest risk where this is most commonly seen in Southeast Asia and Africa [1,2]. As essential micronutrients to humans, these compounds play numerous roles in biological processes, such as vision, gene expression, and immunity [3,4]. The major consequences of VAD include visual impairment, xerophthalmia and night blindness, and VAD significantly increases the risk of severe infections and diseases, and can even result in death. United Nations International Children’s Emergency Fund (UNICEF), in partnership with World Health Organization (WHO), has recommended vitamin A supplementation programs and food fortification strategies to alleviate VAD issues [5,6,7]. Unfortunately, these nutritional interventions have progressed slowly due to a low coverage rate, and there is the persistent concern that delivering excess preformed vitamin A to consumers may result in hypervitaminosis A, which can be lethal or contribute to teratogenic mutations [8]. Thus, a food-based approach, based on ingredients rich in pro-vitamin A carotenoids, seems a more prudent choice, and to that end, this study sought to develop a novel food product using sweet potatoes, peanuts, chickpeas, and natural spices for inclusion in such a program.

Sweet potatoes were chosen as they are an excellent source of bioavailable β-carotene [100–1600 μg retinol activity equivalent (RAE)/100 g for varieties in Africa] [9], which is the most active form of carotenoids without toxicity-based side effects [10]. Furthermore, in recent years, Centro International de la Papa (CIP) (International Potato Center (English)) has initiated several agricultural programs to improve the dissemination of bio-fortified, β-carotene-rich sweet potatoes in sub-Saharan Africa and Asia by coordinating with local governments and implementing advanced production and processing technologies [11].

Peanuts were chosen to provide dietary fat, which in turn enhances the absorption and bioavailability of carotenoids [12], as consumption of carotenoid-rich foods without sufficient dietary fat results in low absorption. Luckily, peanuts are composed of approximately 50% fat [13] and are especially rich in monounsaturated fatty acids (MUFAs), whose absorption and metabolism occurs in the proximal part of small intestine where β-carotene transport proteins are localized and absorption predominately occurs [14,15]. Therefore, the fat from peanuts can largely improve the bioavailability of β-carotene and potentially improve serum vitamin A status.

Finally, chickpeas were added to the base formulation to complete the amino acid profile, making the developed food product more nutritionally complete [16]. While previous work has examined the impact of thermal processing on starch gelatinization of chickpeas [17], it is unlikely that the protein structure is negatively affected by processing. As such, the digestible carbohydrate content may be reduced whereas the protein content is at best unaffected and, at worst, unknown. While the base formulation addressed the nutritional needs, there was some concern surrounding product palatability, and to address this, one of two spice blends were examined: curry spices and pumpkin-pie spices. This combination of ingredients allowed the use of sensory techniques to determine the potential acceptability of the final formulations.

The overall objective of this research was to measure the effect of the differences in formulation on the overall quality of the novel functional food designed to address VAD, particularly as they relate to the effects of flavor and ingredient combination on color, texture and consumer acceptability. It was hypothesized that while color and textural changes may occur due to the added flavorings, the acceptance of the curry-flavored and pumpkin spice-flavored products will be significantly greater than that of the unseasoned product. In the present study, it was determined that a β-carotene rich food with complete protein was well-received by the sensory panel with the greatest liking attributed to curry flavored samples. Furthermore, no differences were found between ethnic groups on flavor preference indicating that it may be acceptable in multiple regions of the world.

## 2. Experimental

### 2.1. Sample Preparation

All ingredients and spices, including sweet potatoes, commercial pure peanut butter, canned chickpeas, ground cinnamon, ground ginger, ground cloves, ground nutmeg, curry powder, cumin powder, turmeric powder, garlic powder, and salt were purchased from a local supermarket (Kroger, Athens, GA, USA). Total vitamin A content (in retinoic acid equivalents (RAE)) was estimated to be 1442 RAE in a 250 g serving based on US Department of Agriculture (USDA) food composition estimates. Initial ingredient formulations were developed using The Food Processor Nutrition and Fitness Software (ESHA Research, version 11.3.285). The ingredient titles used from this software were sweet potatoes, baked in skin, w/salt, mashed (USDA, ESHA code-5994), Peanut butter, creamy (USDA, ESHA code-4627), and Chickpeas, canned, drained (USDA, ESHA code-38880).

Prior to use, raw sweet potatoes were washed with municipal tap water and dried using paper towels. The sweet potatoes were then cut into 1-cm slices with skins by an electric slicer (Model 134054, Hobart Corporation), and placed on perforated stainless-steel boards for steaming. Steaming was then performed to destroy cell structure and reduce enzymatic activities using a commercial steam cabinet (Model SB 24·24, Pyramid Food Processing, Inc., Newark, NJ, USA) at 95 °C for 8 min. Immediately after steaming, potato slices were plunged into tap water for 30 min, or until the temperature of the slices was reduced to at least 33 °C. The cooked sweet potatoes were then processed into a puree using a commercial food processor (RSI 10V, Robot coupe^®^) at 1200 rpm for 1 min. The resultant puree was mixed with pure peanut butter and intact canned, skinned chickpeas at a ratio of 60:35:5 (*wt*/*wt*/*wt*) to form the base formulation. For formulations with added flavor, the respective ingredients were mixed into the base formulation as described in Table 1. Portions (250 ± 5 g) of each mixture were packaged into a multi-layer laminated retort pouch, and thermally processed using a rotary retort simulator (Universal 610-10, FMC Technologies, Inc., Santa Clara, CA, USA) which had been fitted with a custom-built pouch basket. Two pouches in each processing run were fitted with packing glands to accommodate thermocouples to measure internal temperature. Prior to pouch loading, the retort equipment was preheated to 121 °C, and all thermocouples were connected to a data logger. The retort was then sealed and operated at 125 °C. When the internal temperature of both thermocouple-fitted pouches passed 121 °C, the pressure and temperature of the whole system was reduced to 121 °C and maintained for 4 min in order to achieve 12-log reduction sterilization for all products [18]. The system was then gradually cooled down under pressure to avoid pouch rupture, until the internal temperature of the pouches reached 60 °C or below, at which point the system was drained and opened. The processed pouches were then immersed into an ice water bath to limit further cooking. After immersion chilling, these products were stored under refrigeration at 4 °C until the day of analysis.

### 2.2. Analysis of Physical Properties

Tests of firmness were performed on samples which had been allowed to come to room temperature (22 °C). Testing was accomplished using a Texture Analyzer fitted with a 35 kg load cell (TA.XT Plus-upgrade, Texture Technologies Corp. and Stable Micro Systems, Ltd., Hamilton, MA, USA) and a 40 cm diameter polished aluminum cylindrical probe. For each test, 100 g of each sample was loaded on a sample holder and evened out to a uniform 1 cm thickness. The testing speed was 0.83 mm/s, and the samples were compressed to 75% of its height (thickness). Firmness or hardness of each sample was defined as the maximum force required to compress the sample and expressed as N force/g sample, and tests were completed in triplicate.

The color of each formulation was evaluated at room temperature (22 °C) using a colorimeter (Model CR-410, Konica Minolta Sensing Americas Inc., Ramsey, NJ, USA) which was calibrated on the day of analysis according to the manufacturer’s instructions. For each measurement, a portion of each sample was placed in the granular materials attachment and evaluated using the CIE L*a*b* scale. All formulations were measured in triplicate. Water activity and pH were also measured in triplicate at room temperature using a benchtop water activity meter (AquaLab Series 3T, Decagon Devices Inc., Pullman, WA, USA) and a calibrated pH meter (Accumet Basic # 201400.1, Fisher Scientific Inc., Hampton, NH, USA), respectively.

### 2.3. Sensory Evaluation

To measure the acceptability of the overall product, as well as specific attributes, including appearance, flavor, and texture, a 9-point hedonic scale was used. In this scale, 9 points are labeled with both numbers and verbal anchors to facilitate subsequent data analysis and reduce response biases, where 1 = “Dislike extremely,” 2 = “Dislike very much,” 3 = “Dislike moderately,” 4 = “Dislike slightly,” 5 = “Neither like nor dislike,” 6 = “Like slightly,” 7 = “Like moderately,” 8 = “Like very much,” and 9 = “Like extremely” (Stone & Sidel, 2004). Sensory testing was conducted at the Sensory Lab in the Food Processing Laboratory, University of Georgia. Panelists who self-identified as smokers or regular heavy users of alcohol (defined as people who drink more than 5 glasses of wine or beer per day) were excluded from this research due to a high prevalence of smell and taste impairment among these populations [19]. Pregnant or breast-feeding women were also excluded due to the effects of hormone changes that occur during pregnancy, which have been reported to alter the taste and aroma perception of foods and drinks [20].

During the test, participants were presented with a total of three samples of the processed product as indicated in Figure 1, and were asked to rate each sample for overall liking, flavor, and texture. Panelists received only one sample at a time to reduce the likelihood of inter-sample comparison. Samples were encoded with a randomly generated three-digit number, and each sample was portioned to a uniform mass of 20 ± 1 g and placed into individual translucent polystyrene portion cups with lids (Dart Solo “2 oz” model P200N, Mason, MI, USA). Panelists were served samples at room temperature and steamed white rice and distilled water were provided as palate cleansers. Additionally, after completing evaluation of one sample, panelists were given a mandatory 2-min rest in order to minimize any lingering flavors. The presentation order of samples was randomized, and demographic data (age, gender, nationality, occupation, and education background) were also collected to assist with data trend analysis. Ethical oversight for the use of human subjects was provided by the University of Georgia Institutional Review Board.

### 2.4. Statistical Analysis

All analysis of collected data were performed using statistical software (IBM^®^ SPSS^®^ Statistics v 25.0, IBM corp., Armonk, NY, USA). Physical and sensory properties of the three samples were tested via one-way analysis of variance (ANOVA), with post hoc Tukey honest significant difference (HSD) tests performed as appropriate. A Levene’s test was also conducted to test the homogeneity of variance. To determine the influence of nationality/country of origin on the acceptability of the three products, nationality/country of origin groups were combined along continental lines. For example, the panelists identifying as either US nationals or American-born Chinese were combined and formed the “US” group; Panelists identifying as Chinese, South Korean, Indian, Japanese, Nepalese, or Iranian were combined into the “Asian group”. A two-way ANOVA was then conducted to test if statistically significant differences existed between the groups. For all statistical tests, results were considered significantly different if *p* ≤ 0.05.

## 3. Results and Discussion

### 3.1. Physical Properties

Water activity, pH, color (CIE Lab), and hardness values of the three product formulations (unflavored sample, curry-flavored sample, and pumpkin spice-flavored sample) are presented in Table 2. Results are expressed as mean ± standard error (n = 3). Water activity showed no statistical difference between formulations while the pH values of the unflavored sample was significantly higher than both the curry-flavored sample (*p* = 0.02) and the pumpkin spice-flavored sample (*p* = 0.04). This may influence consumer perception of the product, as all three mean pH values were less than 7, and acidic foods are typically perceived as slightly sour, and since differences in pH as low as 0.05 units have been found to be detectable by taste, both flavored samples may taste more sour than the unflavored sample [21].

Regarding color, all of the sample’s measurements were statistically different, with every sample predictably falling into the red, and yellow categories of a* and b*. The lightness of the unflavored and curry flavored samples were both above the midpoint for L*, indicating those two could be described as “light” or “bright”, but the pumpkin spice sample, probably due to the cloves and cinnamon, had a L* value below 50 (39.0 ± 0.4), which would likely be described as “dark”. Comparing the color measurements between formulations, the unflavored formulation was the lightest (*p* = 0.001) amongst all three formulations, while pumpkin spice-flavored sample was both the most red and least yellow among all three samples (*p* < 0.001).

The hardness of the formulations showed an interesting trend. The unflavored and curry-flavored formulations were statistically similar (*p* = 0.5) as were the pumpkin spice and curry-flavored samples (*p* = 0.09) However, the hardness of pumpkin spice-flavored formulation was significantly greater than the hardness of unflavored sample (*p* = 0.02), possibly due to interaction between the spices and other ingredients in the former. Of particular note is cinnamon, which is quite hygroscopic, and research has shown that it can result in a significant increase in the hardness of breads [22].

Color variations amongst the three samples were also likely due to the addition of spices, as spices have a long history of use as food additives and/or preservatives to give or improve food color, flavor, texture, aroma, and other sensory properties. Natural spices, like those used in this study, can impart significantly more complex color and flavor to foods due to their physiochemical properties and the phenols, flavonoids, and terpenes also give unique properties to all spices [23,24].

In the current research project, the carotenoids from the sweet potatoes were almost certainly responsible for the reddish orange color of all product formulations [25]. Furthermore, the flavonoids native to ground ginger, cumin powder, and turmeric powder are likely responsible for the increase in yellowness in the curry-flavored product [25]. In the pumpkin spice-flavored sample, the reason for the darker, more brown color is probably a combination of the cinnamaldehyde and high levels of phenolic compounds in the ground cinnamon and ground cloves [26,27].

### 3.2. Analysis of Acceptability

The total number of panelists with valid results (all three samples evaluated, no omissions) was 105; 35 male and 70 female. Mean scores for overall liking, flavor, texture, and appearance for all three samples are expressed as mean ± standard error and listed in Table 3. Hedonic ratings for overall liking, flavor, and texture were significantly different amongst the three formulations, with the curry-flavored formulation having the highest score in all three, while pumpkin spice-flavored formulation had the lowest. Ratings for appearance showed no difference in consumer opinion between the unflavored and curry-flavored formulations, both of which were significantly higher than the pumpkin spice formulation. These scores suggest that participants tended to prefer the curry-flavored formulation out of the three. However, the unflavored formulation’s ratings ranged from “Neither like nor dislike” to “Like slightly,” suggesting that some additional flavoring is needed for wide acceptance. Overall, all ratings for all samples fell within the middle of the 9-point scale, which may be due to the central tendency bias, that is, participants often are less willing to choose extreme choices, although their true feelings might have been more towards the extreme options [28]. With the lowest mean score, pumpkin spice-flavored formulation ranged from “Dislike slightly” to “Dislike moderately” in both overall likening and flavor, indicating a need for reformulation or abandonment of that formula to better ensure consumption by the target market. The appearance of the products may also need some attention, as the plain and curry-flavored samples had mean scores of 5.3 and 5.4, respectively, indicating participants “Neither like nor dislike” their appearance. The pumpkin spice formulation had a mean score of 4.3, suggesting participants slightly disliked its appearance, likely due to the dark color of the sample. Panelists were given space on the ballot to provide comments, and from those comments, several panelists indicated that a sweeter or saltier taste was expected when participants evaluated the pumpkin spice-flavored sample, which may have caused an error of expectation. Participants, particularly US participants, who were familiar with the “typical” flavor of pumpkin spice products, may have a fair amount of expectation/anticipation about what was tested, even though the sensory tests were completely blind. This could then cause them to make inferences and bias their judgements due to the product not meeting their expectations [28]. In particular, there was no added sugar to the product to make the product sweet, in contrast to the majority of pumpkin spice products in the US.

The different acceptability of three products resulted from the different flavor, texture, and appearance of the products, which were in turn influenced by the chemical compounds responsible for the specific flavor and aroma of the added flavor ingredients [27]. For example, it is likely that people preferred the flavor of the curry-flavored sample due to the increases in palatability brought to this puree by the salt, which would balance the modest sweet taste of sweet potato and the aroma imparted by the curry powder and cumin spices. In addition, the aromatic clues may have led panelists to anticipate a savory, rather than sweet product, unlike the pumpkin spice formula which likely did the opposite.

Examining the interplay between the physical and sensory properties of the three formulations, some trends can be extrapolated. Specifically, a relatively light yellow and orange color was desirable for puree products, and that consumers preferred a product which was neither too hard nor too soft. There may be some concern that the participants were untrained panelists and not the actual target audience (i.e., US residents and not sub-Saharan African or Asian participants) which may bias the results and prevent their extrapolation to the target consumers [29]. As previously mentioned, the proposed food product was designed for people suffering from vitamin A deficiency, especially children in sub-Sahara African areas, and access to such panelists was unfeasible. Despite that, the authors realize that there are cultural effects on sensory preferences and therefore, the effect of demographics on the sensory insights of the three product formulations was analyzed in the following context.

Consumer acceptability of a product can be influenced not only by sensory inputs, but also by factors related to demographics. The differences in diet and food ingredient use between peoples of different cultures not only impacts the type of foods that people are likely to eat, but also their opinions of certain taste modalities, responses to sensory stimuli, etc. [30,31]. In the current study, a diverse sensory panel was used which allowed for analysis of the potential effect of demographics. Sensory scores were analyzed by ethnicity/nationality groups to determine how uniform the acceptability response was across panelists of differing background. Full demographic data is presented in Table 4, and Table 5 presents the sensory data from Table 3, broken down by the nationality groups mentioned above. The demographic distribution among 105 participants was diverse, though the individual sample sizes (n) for all but two nationalities (US and China) were relatively small within populations. Therefore, we combined demographics into regional or continental groups as a more appropriate categorical method. The results show that the demographic effect was negligible and that there was no significant difference between various demographic groups (*p* > 0.1) This suggests that the products would be expected to have similar levels of acceptability across cultures, which is consistent with previous studies [32,33], and which would be ideal for a product designed to address worldwide nutritional concerns.

The negligible demographic effect may be due to rapid globalization. Cultural transference, especially food globalization, is occurring particularly in the US and China, where nearly 80% of sensory panelists were originally from. Therefore, the majority of participants were likely experiencing taste convergence [34], which may bias the results of the current study. However, as a preliminary sensory analysis, the data suggest acceptability of the product and provide a preliminary framework if/when the product is assessed in the target region(s). There is reason to assume that the product’s acceptability would be relatively high in Africa, as the past 30 years have seen an increase in food imports across the region which would contribute to taste convergence [35]. Therefore, the lack of a demographic effect on consumer score should be seen as a positive outcome wherein the products may be accepted by the target populations and could be used as a food-based approach to combat VAD in those countries.

The results of this research show that the curry-flavored formulation was the best candidate of the three, due to it receiving the highest hedonic scores regarding its overall liking, flavor, texture, and appearance among all three samples. It further suggests that it is possible to create an acceptable product to address vitamin-A deficiency using commodities available in the target locations. Furthermore, the use of pro-vitamin A carotenoids reduces the risk of costly and potentially dangerous supplements. The high acceptability of the curry-flavored formulation was likely due to the use of salt and spices, which were able to improve the texture, flavor, and color of the unflavored formulation. Moreover, since there was no significant geographical or age effect on product preference, it is highly likely that the curry-flavored formulation could be implemented in low-income countries and would be accepted by people of all ages. Despite the positive outcomes, there are several limitations to this research. Future studies would be needed to conduct a series of trials among the target population, as would recreating the product using the versions of the ingredients and materials available in the target areas (i.e., orange-fleshed sweet potato versus US sweet potatoes). In addition, optimal study design considerations would be preferred versus sub-optimal design methods [36,37]. Despite these limitations, the results from this research can provide useful preliminary sensory data and a solid theoretical foundation for the development of food-based approaches to combat VAD.

## Figures and Tables

**Figure 1 foods-09-00232-f001:**
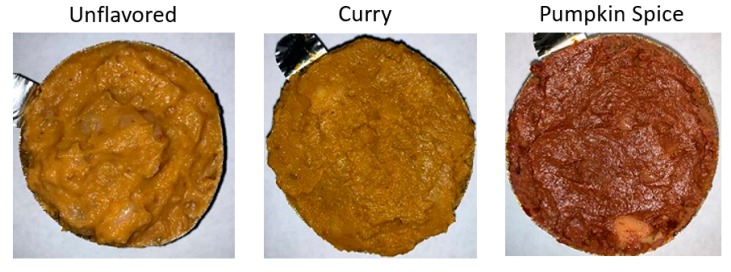
Representative images of the three samples used in the sensory and physical property studies. Samples were prepared as indicated and served to participants as illustrated in the figure in a randomized order. Sensory data from an unflavored sample was compared to a curry-flavored and a pumpkin-spice flavored sample. Physical properties were also determined and presented in the analyses.

**Table 1 foods-09-00232-t001:** Product formulations for each of the three products used in the studies.

Formulation	Flavor	Formulation	
1	Unflavored Sample	Sweet potato puree	60 g
		Pure peanut butter	35 g
		Peeled chickpeas	5 g
2	Curry-Flavored Sample	Salt	1.6 g
		Curry powder	0.6 g
		Ground cumin	0.6 g
		Ground turmeric	0.48 g
		Garlic powder	0.28 g
		Unflavored formulation	100 g
3	Pumpkin Spice-Flavored Sample	Ground cinnamon	0.61 g
		Ground cloves	0.38 g
		Ground nutmeg	0.16 g
		Unflavored formulation	100 g

**Table 2 foods-09-00232-t002:** Physical properties of the three formulations of products used in the studies.

Formulation	a_w_	pH	Color			Texture (Hardness) N/*g*
			L*	a*	b*	
Unflavored	0.98 ± 0.01 a	4.9 ± 0.06 a	54.4 ± 0.9 a	8.8 ± 0.4 a	38.7 ± 0.9 a	9.7 ± 4 a
Curry-Flavored	0.96 ± 0.01 a	4.5 ± 0.1 b	50.5 ± 0.2 b	6.5 ± 0.2 b	38.5 ± 0.4 a	16.4 ± 5 ab
Pumpkin Spice-Flavored	0.98 ± 0.01 a	4.6 ± 0.04 b	39.0 ± 0.4 s	13.2 ± 0.03 c	14.2 ± 0.7 b	25.1 ± 6 b

Mean values, ± standard error (SE, n = 3); different letters within the same column indicate significant different at *p* ≤ 0.05 (2-sided).

**Table 3 foods-09-00232-t003:** Mean scores for attributes of the three formulations of products used in the studies using a 9-point hedonic scale.

Formulation	Overall Liking	Flavor	Texture	Appearance
Unflavored	5.6 ± 0.2 a	5.6 ± 0.2 a	5.6 ± 0.2 a	5.3 ± 0.2 a
Curry-Flavored	6.7 ± 0.2 b	6.8 ± 0.2 b	6.2 ± 0.2 b	5.4 ± 0.2 a
Pumpkin Spice-Flavored	3.8 ± 0.1 c	3.8 ± 0.2 c	5.0 ± 0.2 c	4.3 ± 0.2 b

Mean values, ± SE, (n = 105); different letters within the same column indicate significant different at *p* ≤ 0.05 (2-sided).

**Table 4 foods-09-00232-t004:** Demographic information of participants.

Demographics	Category	Sample Size *n* = 105
Gender *n* (%)	Female	70 (66.67%)
	Male	35 (33.33%)
Age *n* (%)	18–25	64 (60.95%)
	26–35	32 (30.48%)
	36–45	5 (4.76%)
	46–55	1 (0.95%)
	Above 56	3 (2.86%)
Occupational Status *n* (%)	Student	83 (79.05%)
	Employed for wages	22 (20.95%)
Education Background *n* (%)	High school graduate	10 (9.52%)
	College, no diploma	28 (26.67%)
	Bachelor’s degree	28 (26.67%)
	Master’s degree	28 (26.67%)
	Doctorate degree	11 (10.48%)
Nationality *n* (%)	US	60 (57.14%)
	American-born Chinese	1 (0.95%)
	China	25 (23.81%)
	South Korea	6 (5.71%)
	India	3 (2.86%)
	Japan	2 (1.90%)
	Iran	2 (1.90%)
	Nepal	2 (1.90%)
	Turkey	2 (1.90%)
	Brazil	1 (0.95%)
	Haiti	1 (0.95%)
Vegan/Vegetarian *n* (%)	Not vegan/vegetarian	97 (92.38%)
	Vegan/vegetarian	8 (7.62%)

**Table 5 foods-09-00232-t005:** Mean scores for attributes of the three formulations of products used in the studies using a 9-point hedonic scale.

	Overall Liking	Flavor	Texture	Appearance
*Unflavored*				
US	5.5 ± 0.2 a	5.6 ± 0.2 a	5.5 ± 0.2 a	5.2 ± 0.2 a
Asian	5.8 ± 0.3 a	5.8 ± 0.3 a	5.8 ± 0.3 a	5.5 ± 0.3 a
*Curry-Flavored*				
US	6.7 ± 0.2 a	6.9 ± 0.2 a	6.1 ± 0.2 a	5.5 ± 0.2 a
Asian	6.7 ± 0.3 a	6.8 ± 0.3 a	6.4 ± 0.2 a	5.2 ± 0.3 a
*Pumpkin Spice-Flavored*				
US	3.7 ± 0.2 a	3.9 ± 0.2 a	5.0 ± 0.2 a	4.2 ± 0.2 a
Asian	4.0 ± 0.3 a	3.9 ± 0.3 a	5.0 ± 0.3 a	4.4 ± 0.3 a

Mean values, ± SE; different letters within the same column indicate significant different at *p* ≤ 0.05 (2-sided). US: n = 61, Asian: n = 44.

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
