# Peer review of "Sensory Analysis of a Processed Food Intended for Vitamin A Supplementation"

_foods, 2020, doi:10.3390/foods9020232_

Round 1
Reviewer 1 Report
I like the paper, it is well written and the content is required. The gap between nutritional reformulation and product fulfilment is one that needs to be closed. I have made the authors aware of a minor typo and a reference that I think may be useful. There is a suite of work on pulses investigating bioavailability of proteins and this might be worth highlighting here. I prefer sensory results to be graphed rather than tabulated for ease of presentation but this is a personal preference. I think the authors could summarise a conclusion for the study.
However, the paper is well written and I believe of considerable importance, it begins to link bioavailability to reformulation and this is a major requirement for food and beverage industries worldwide.
Check reference citing formats, is this journal style? I think there is a reference manager style sheet available for this.
See file attached, minor comments, authors can consider the above, I think a conclusion paragraph will improve impact of the paper.

Author Response
We would like to thank reviewer 1 for their time and attention to the manuscript. The comments and suggestions are welcomed and we have made every attempt to address them and our responses to each point are listed below:
1) Graphed sensory results vs. tables: We agree that graphed data are generally much easier to interpret and as a reader, it is much easier to follow. However, in the present study, we compared the data as graphs vs. tables and we found that the latter was less confusing. Since we have three formulations and multiple outcome measures, the table format seemed to summarize that data and statistics in a more concise manner. The graphs became a little excessive and 'busy'. We would prefer to keep the data in the table form, however, if reviewer 1 wold prefer it in graphical format we would be happy to accommodate.
2) Provide a summary of the conclusion: We have provided a brief, two-sentence summary at the end of the introduction.
3) References were corrected to match MDPI style.
4) The information from Edwards et. al has been included in the text. We agree with reviewer 1 that it is an important consideration and we should have included some consideration of the effect of thermal processing on macronutrient digestibility. As such, we have acknowledged this fact in the text.
Reviewer 2 Report
Introduction
A brief introduction of papers on reformulation is required here. Some references may be useful.
It remains unclear to me what the authors are aiming for, as such a novel functional food is developed - but what is it and where it will sit as a food product isn't explained. More information is needed.
Method
Line 146 - it's CIE L*a*b* instead of CIEL*a*b*
Sensory analysis
How was the sample presented? Just as paste?
Stats Analysis
On nationality stats, why is one way ANOVA performed here? I would recommend 2 way ANOVA where the nationality is set as an effect together with the product. As such interactions should also be considered here - as such the results for acceptability section is questionable especially on the segmentation between US and Asian consumers.
Results
Line 242 - Hednonic to hedonic
Line 306 - For consistency China to Asian? USA to US?
As such the n.s. between demographic groups may also derive from the lack of consumer numbers in the study - the presented study only recruited 105 participants which may be statistically underpowered when segments are identified. I'd recommend the authors to conduct cluster analysis instead of segmenting the consumer pools into different nationalities. This will perhaps bring more insights to the data that has been gathered.
Fig 1 caption needs to be revised, more information is needed, what was the samples presented?
Table 1 caption, product formulation of what? More details is required. A caption should have sufficient information as a standalone.
Table 2 caption, same as above - what three formulation was it? Column L* for pumpkin-spice has 's' as a grouping which I assume is a mistake. 2 sided significance, is it a posthoc comparison (e.g. Tukey), if yes - please add more info.
Table 3 caption, reformulation of..?
Table 5 caption, 'US people, Asians' - needs to be rephrased to US population and Asian population? Or ethnicity? People doesn't make sense here. All the groupings are 'a', therefore no significant differences? If that's the case then perhaps delete the groupings and add an additional caption indicating no significance were reached.
Conclusion
Limitations of the study isn't stated here, as such a better design of experiment can be used, as such a d-optimal design - see ref for inspiration
Gao, Y., Hamid, N., Gutierrez-Maddox, N., Kantono, K., & Kitundu, E. (2019). Development of a probiotic beverage using breadfruit flour as a substrate. Foods, 8(6), 214.
Or conducting a descriptive analysis and temporal methods on sensory percept would be beneficial to provide a whole picture for profiling
Kantono, K., Hamid, N., Oey, I., Wang, S., Xu, Y., Ma, Q., ... & Farouk, M. (2019). Physicochemical and sensory properties of beef muscles after Pulsed Electric Field processing. Food Research International, 121, 1-11.
Author Response
We would like to thank Reviewer #2 for their time and attention to our manuscript. Below are the point-by-point responses:
1) 'A brief introduction of papers on reformulation is required here:' As we understand the topic of reformulation, it refers to refinements and improvements on existing products that are intended to reduce certain micro- or macronutrients (e.g. sodium or sugar). This was not the purpose of the present study- in fact, the is an entirely novel product that we developed and subsequently assessed the sensory and physical properties. We did not alter an existing product and then determine the changes relative to the original - which as we understand it, would fall under the category of reformulation. However, if we are mistaken in our understanding, we would be happy to revisit this topic.
2) CIEL*ab has been changed to CIE L*a*b*
3) How was the sample presented? Just as paste?: That is correct, the samples were presented as illustrated in Figure 1. We realized that we had failed to reference Figure 1 in the text but that is now added for clarification.
4) 1-way versus 2-way ANOVA- Reviewer #2 is correct, we should have included the statement that we used both a 1-way and 2-way ANOVA in our statistical analyses. This additional We chose to use a 1-way ANOVA to determine if there were any differences in preference between the three formulations independent of ethnicity. We later assessed ethnic effect, and that was a 2-way ANOVA and that was corrected in the text. There were no significant interactions observed in the analysis.
5) Hednonic was changed to Hedonic
6) We changed all appropriate mentions of 'Chinese' to 'Asian' and 'USA' to 'US'.
7) We are very sorry that we missed the inclusion of the 'other' group from Table 5. The only two groups that were intended to be in that analysis were US-born and non-US born Asians. Thus, with a third group of 'other' with an n=4, that is severely underpowered. However, those subjects were not intended to be in the analysis for Table 5 and have been removed from the text.luster analysis and ns results: Reviewer #2 raises and interesting point regarding the use of cluster analysis for the sensory data. While this point is certainly valid in the context of multiple groupings of nationalities, we do not have sufficient power in most, if not all individual groups. Therefore, participants were grouped into US and non-US born in order to determine if that had an impact on preference.
8) Figure 1 caption has been expended.
9) We are not sure where the confusion exists concerning the table legends and the data contained in the tables. When the legend indicates 'the three formulations', we assumed it was clear that the table data corresponds to the table legend. Unlike figures, tables generally to no have extensive descriptions as the data in the table are explained by each heading/category/column within. However, as per Reviewer #2's request, more information has been provided in the table listings to make this clearer.
10) Table 2- Since we only used three formulations in the study, this refers to the three formulations we used in the study and indicated in the table. No others were ever presented, discussed, or listed thus we do not understand where the confusion is coming from. If we have somehow misled Reviewer #2 into thinking there were more than 3 formulations, we are very sorry for this misunderstanding. If the point of confusion could please be provided to us, we would be very happy to correct it.
11) Table 3: We do not list 'reformulation', we only discuss formulation.
12) Table 5 caption has been corrected.
13) We have included an additional consideration of study design in the study limitations.
Round 2
Reviewer 2 Report
i'd like to thank the authors for providing the revisions needed for the manuscript.
A minor comment that seems to be misinterpreted is on the last response re sub-optimal design is to use proper design of experiments (DoE) such as response surface method to optimise the recipe further. As such following the below paper that utilised mixture design.
Gao, Y., Hamid, N., Gutierrez-Maddox, N., Kantono, K., & Kitundu, E. (2019). Development of a probiotic beverage using breadfruit flour as a substrate. Foods, 8(6), 214.
Author Response
We would like to thank Reviewer #2 for their time and attention to our manuscript.